# Crystal structures of the elusive *Rhizobium etli* L-asparaginase reveal a peculiar active site

Joanna I. Loch[1], Barbara Imiolczyk[2], Joanna Sliwiak[2], Anna Wantuch[1], Magdalena Bejger [2], Miroslaw Gilski [2,3] & Mariusz Jaskolski [2,3✉]

*Rhizobium etli*, a nitrogen-fixing bacterial symbiont of legume plants, encodes an essential L-asparaginase (ReAV) with no sequence homology to known enzymes with this activity. High-resolution crystal structures of ReAV show indeed a structurally distinct, dimeric enzyme, with some resemblance to glutaminases and β-lactamases. However, ReAV has no glutaminase or lactamase activity, and at pH 9 its allosteric asparaginase activity is relatively high, with $K_m$ for L-Asn at 4.2 mM and $k_{cat}$ of 438 s$^{-1}$. The active site of ReAV, deduced from structural comparisons and confirmed by mutagenesis experiments, contains a highly specific $Zn^{2+}$ binding site without a catalytic role. The extensive active site includes residues with unusual chemical properties. There are two Ser-Lys tandems, all connected through a network of H-bonds to the Zn center, and three tightly bound water molecules near Ser48, which clearly indicate the catalytic nucleophile.

[1] Department of Crystal Chemistry and Crystal Physics, Faculty of Chemistry, Jagiellonian University, Krakow, Poland. [2] Institute of Bioorganic Chemistry, Polish Academy of Sciences, Poznan, Poland. [3] Department of Crystallography, Faculty of Chemistry, A. Mickiewicz University, Poznan, Poland. ✉email: mariuszj@amu.edu.pl

L-Asparaginases, which split L-Asn to L-Asp and ammonia, are a diversified group of enzymes[1]. The serendipitous detection of strong antileukemic properties of guinea pig serum[2] was attributed by Broom et al.[3] to L-asparaginase activity, and indeed low (μM) $K_m$ bacterial asparaginases (Class 1 in revised classification[4]) of type II[5], notably *E. coli* EcAII, but not type I (e.g. EcAI with mM $K_m$[6]) became potent drugs in the treatment of acute lymphoblastic leukemia (ALL)[7]. Later, it was realized that plants possess different, Ntn-type, asparaginases (some of which, e.g. in legumes, are K$^+$ dependent[8,9]) but close homologs of plant-type (or Class 2) asparaginases were also discovered in bacteria, e.g. EcAIII[10,11].

Experiments with *Rhizobium etli*, a nitrogen fixing bacterial symbiont of legumes, revealed that it possesses two asparaginases, a constitutive and an inducible one[12]. A comprehensive phylogenetic analysis of L-asparaginase sequences[13] predicted early on that the inducible *R. etli* enzyme is not related to any known Class 1 or Class 2 asparaginases. According to a recently proposed classification[1] the *R. etli* enzymes are termed ReAIV (constitutive) and ReAV (inducible) and belong to Class 3 asparaginases. An in silico model of ReAV predicted a protein fold with structural similarity to β-lactamases or transpeptidases[14].

Because of serious side effects of bacterial asparaginase administration, other candidate antileukemics have been sought, so far without success, e.g. by mutagenetic optimization of natural enzymes[15] or by considering plant-type mammalian proteins in this role[16–18]. Also, interest has been directed towards *R. etli* asparaginases, which could offer at least a possibility for optimization. The structure of these enzymes has been, however, recalcitrant to experimental elucidation; for example, in our laboratory futile crystallographic studies have taken almost 20 years. In this work we finally describe high resolution crystal structure of ReAV in several crystal forms and show that the enzyme belongs to a folding class that is more closely related to glutaminases or penicillin-binding proteins than to any of the known L-asparaginase families. We report that ReAV is an allosteric asparaginase with high pH optimum and with $K_m$ of 4.2 mM at pH 9, and that it is a metalloprotein with a puzzling active site.

## Results

### Expression and biophysical characterization of WT ReAV and its mutants

WT ReAV was expressed in *E. coli* in large amounts in soluble form (~100 mg/L of bacterial culture). The correctness of the WT ReAV sequence was confirmed by mass spectrometry. Based on analysis of the crystal structures of the WT protein, five ReAV mutants with substitutions in the putative active site area were generated (Supplementary Table 1). A similar expression yield was observed for mutant K51A, and a slightly lower yield for K263A. On the other hand, mutants S48A and S80A were produced mostly in insoluble form as inclusion bodies (Supplementary Fig. 1). Nevertheless, it was possible to recover from the cell lysate 1–5 mg of each variant for routine activity and stability tests.

Poor signal in the far-UV CD spectra indicated that variants S48A and S80A are incorrectly folded or carry a strong structural distortion (Supplementary Fig. 2). These observations were confirmed by nanoDSF experiments, as it was impossible to determine a reliable $T_m$ for the S48A and S80A mutants due to poor fluorescence. $T_m$ determined for WT ReAV is 51 °C (nanoDSF), in agreement with previous results[12,19]. The $T_m$ was similar for variant K51A (50 °C), and slightly higher for K263A (52 °C) or lower for C135A (48.5 °C) (Supplementary Fig. 2). A series of DSC experiments performed for WT ReAV in buffers of different pH and NaCl content showed a two-step melting

process at pH 6.5 and 7.5 and only one transition at pH 8.5. At pH 7.5 in the presence of 10 mM NaCl, the second peak was significantly reduced (Supplementary Fig. 3).

### Crystal forms of ReAV

In the course of the crystallization experiments, a number of different crystal forms of WT ReAV were obtained, four of which are presented herein as follows (Supplementary Table 2): orthorhombic form OP (the highest resolution, 1.29 Å, of X-ray diffraction) with one ReAV dimer in the asymmetric unit (ASU) and tight packing of the molecules; monoclinic form MC with one dimer in ASU; and another monoclinic form with four protein chains (two dimers) in ASU, reported here as two variants, MP1 and MP2, differing in the hydration pattern of Ser48.

The orthorhombic (OP) and monoclinic MP1/MP2 forms show similar packing pattern with solvent content of 49.4–52.6%, whereas the monoclinic form MC, with a significantly different arrangement of the dimers in the unit cell, has 53.2% solvent content.

In all cases the electron density maps have excellent definition for almost the entire protein molecules, except short disordered regions at the N- and C-termini. Even a preliminary analysis of the crystal structures clearly suggested that the active site is located in the area with an accumulation of residues showing unusual behavior, centered around the hydrated Ser48 side chain, and accentuated by a nearby Zn$^{2+}$ ion coordinated by Cys135, Lys138 and Cys189.

### Protein fold topology

The ReAV protomer has an α/β fold formed by two tightly packed domains. The larger, mostly α-helical catalytic domain built of helices H2–H14, harbors the active site (Fig. 1). This domain also contains a small three-stranded β-sheet (strands B4–B6). The second (dimer stabilization) domain consist of a six-stranded antiparallel β-sheet (strands B1–B3 and B7–B9) located in the core of the molecule. This β-sheet also includes a very long and unique β-strand B2 involved together with helix H7 in dimer stabilization.

### The oligomeric state and location of the active site

ReAV is a homodimer with approximate $C_2$ symmetry (Fig. 2). This symmetry is preserved quite well, as illustrated by the ~180° rotation and Cα rmsd of ~0.5 Å between the subunits. The shape of the whole dimer resembles a basket, with the handle formed by the pair of helices H7 from both subunits. The bottom of the basket is stabilized by an extended network of H-bonds and hydrophobic interactions formed between the longest β-strand B2, strand B1 and the extended loop connecting helix H19 and strand B10 from one subunit, and the residues located in strands B4, B5 and B6 from the other subunit (Fig. 2).

There is a characteristic tunnel with ~20 Å diameter between the subunits, suggesting that ReAV might interact with other biomolecules. The tunnel leads to the two active sites (Fig. 3), each located in a cleft on the protomer surface. Calculations of the electrostatic potential revealed strong positive charge in the cleft area that would be excellent for attracting negatively charged ligands/substrates (Fig. 3d). This analysis clearly confirmed that the putative active site is located in the middle of the α-helical domain, in the cleft surrounded by helices H2, H4, H9 and the loop connecting strands B5-B6, in agreement with the conclusion drawn from the chemical character of the residues in this region. This finding has been subsequently confirmed by site-directed mutagenesis in this area, coupled with enzymatic tests. In addition to the unusual arrangement of the putative catalytic residues, the active site cleft is occupied in its distal area by a zinc ion with a singular coordination sphere.

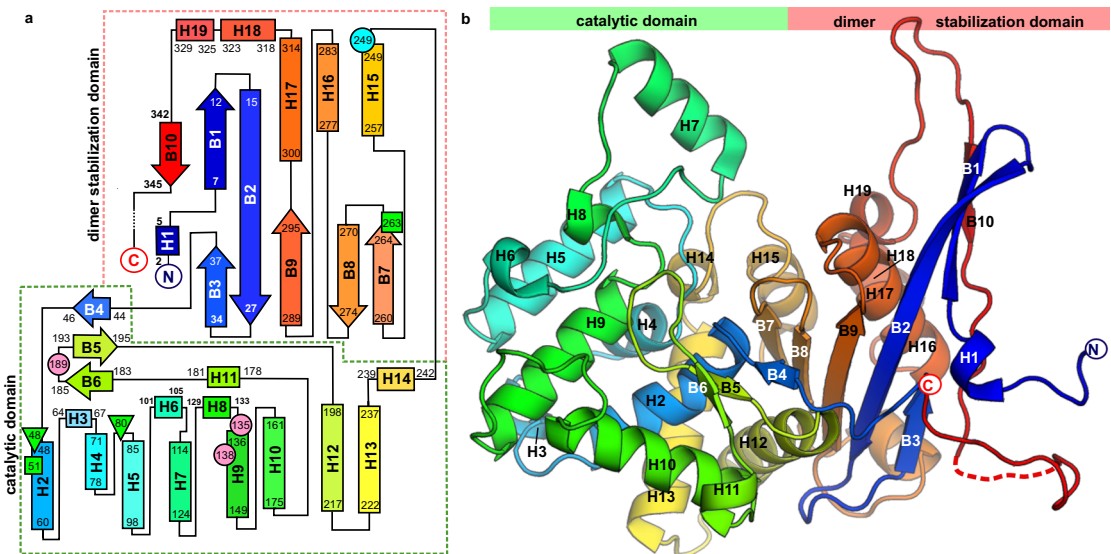

**Fig. 1 Topology of the ReAV chain and overall structure of a single subunit.** In the topology diagram (**a**), residues involved in zinc coordination (Cys135, Lys138, Cys189) are marked as pink circles, while residues potentially involved in catalysis are marked by green triangles (Ser48, Ser80) and green squares (Lys51, Lys263). Cysteine 249, carrying a posttranslational modification, is marked by a cyan circle. The overall fold of a single subunit is shown in (**b**).

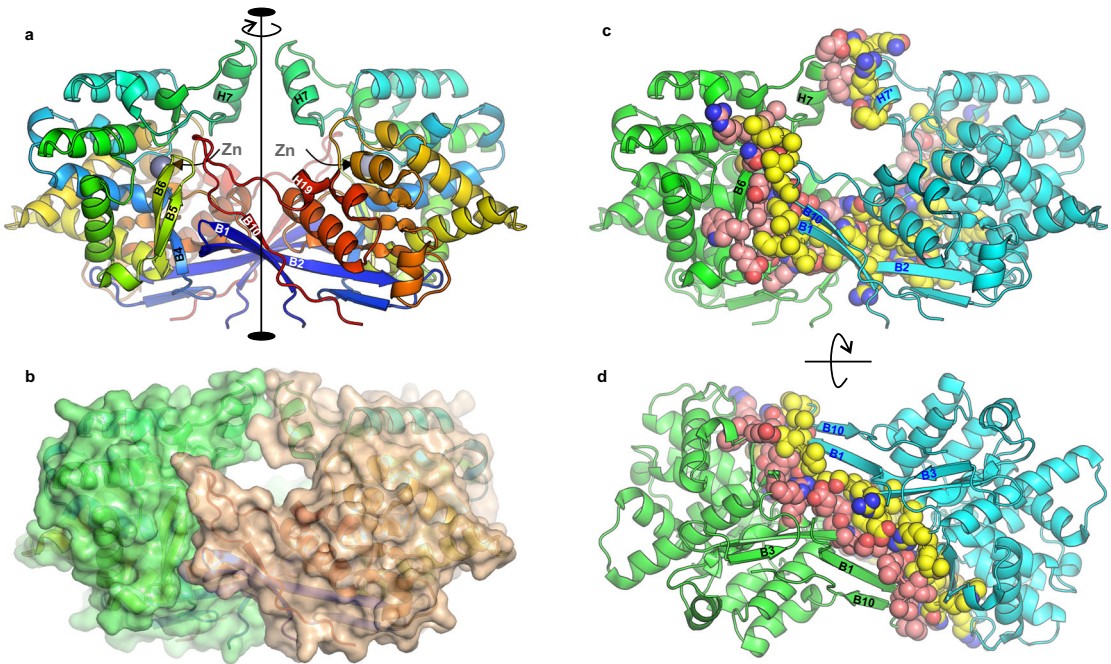

**Fig. 2 The dimeric assembly of ReAV. a** In the $C_2$ dimer, the active sites are located in the center of each subunit and contain $Zn^{2+}$ cations. **b** Molecular surface of the dimer, showing a tunnel of ~20 Å diameter. **c** Residues forming the dimer stabilization domain and helices H7 are marked as spheres. **d** A perpendicular view showing that the area involved in dimer stabilization extends across the dimer to include many hydrogen bonds formed between main-chain and side-chain atoms.

**Metal binding site**. The zinc ion was found near helix H9 and the loop connecting strands B5–B6, with a tetrahedral coordination sphere formed by Cys135, Lys138, Cys189 and a water molecule (w1) (Fig. 3a). The position of Lys138 is stabilized by two water-mediated H-bonds to the side chain of Gln54 (Fig. 3b). These water molecules (w7, w8) and the position of the Lys138 side chain are conserved in all analyzed ReAV structures. The position of Cys189 is nearly identical in all the structures and is additionally held in place by an H-bond to the side chain of His139. Cys135 interacts not only with the Zn ion but also with Lys51, and weakly with Asn134. The two Zn-S distances are almost identical (~2.30–2.31 Å). The distance between the water

molecule w1 and $Zn^{2+}$ is in the range ~1.89–2.05 Å and the position of this water is the same in all structures. The coordination geometry together with an X-ray fluorescence scan and anomalous map, plus ITC titration results (vide infra), all unambiguously identify the metal cation as $Zn^{2+}$.

**The catalytic center**. Close to the $Zn^{2+}$ site, a system of four conspicuous residues, Ser48, Ser80, Lys51 and Lys263, is located. In the close vicinity of the Ser48 hydroxyl, there is a characteristic, almost tetrahedral pattern of three closely-spaced water molecules w2, w3, w4 (Fig. 3e), visible in all crystal structures, except MP2.

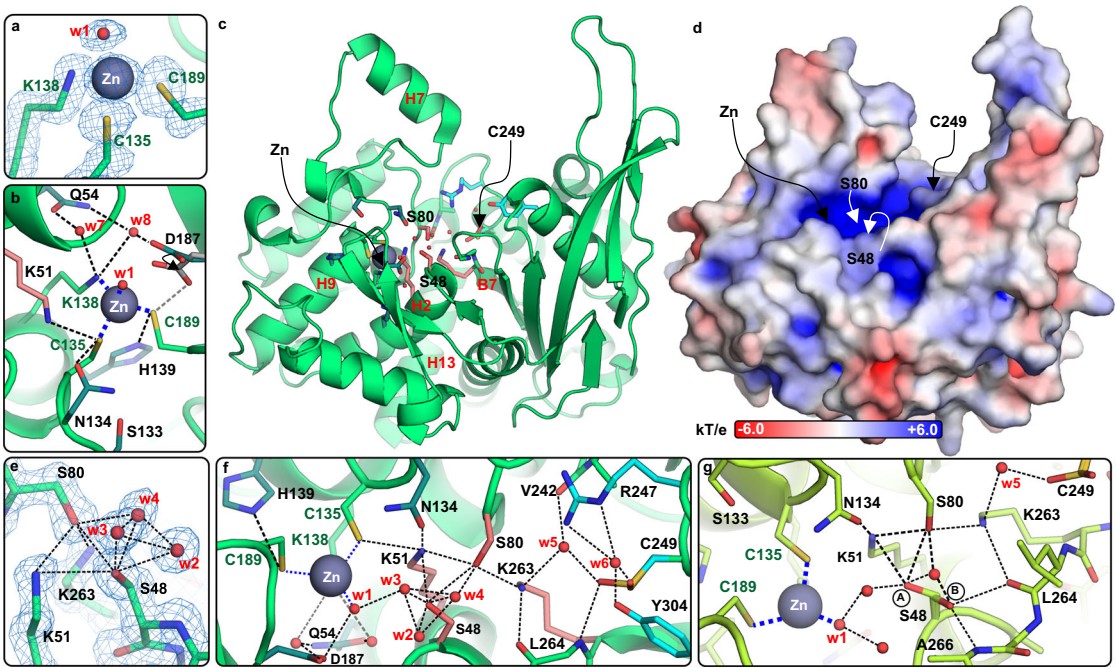

**Fig. 3 Detailed view of the active site area.** Panels **a–f** correspond to the highest resolution (1.29 Å) structure OP, while (**g**) shows structure MP2. **a** 2Fo-Fc electron density (contour level 1.50σ) at the Zn coordination sphere. **b** Hydrogen bonds (black dashed lines) involving residues participating in Zn coordination (thick dotted blue lines); alternative conformations of D187 were visible only in the OP structure. **c** Overall structure of the ReAV protomer with residues creating the active site in stick representation. **d** Surface electrostatic potential with the strongest positive charge (blue) visible in the active site area. **e** 2Fo-Fc electron density (contour level 1.50σ) around Ser48 and the water triad w2-w3-w4. **f** A detailed map of the most important hydrogen bonds in the active site. **g** Ser48 observed in two conformations (circled A and B) in structure MP2, where the water triad is absent.

These waters must have extremely strong H-bonds with the Oγ atom of Ser48, as indicated by the O···O distances, which in the highest-resolution structure OP refined to 2.2–2.5 Å. The water molecules are also tightly H-bonded to each other. Water w3 is also H-bonded to w1 coordinated by the $Zn^{2+}$ cation (Fig. 3f).

In all structures, regardless of the hydration state of Ser48, the side chain of Ser48 is also H-bonded to the Nζ atom of Lys51 and Oγ of Ser80. On the other hand, Ser48 is not always hydrated (structure MP2) or the hydration may be incomplete and even different in the same crystal structure (triclinic crystals, not included in this study). In structure MP2, the side chain of Ser48 has two conformations: one as in the hydrated state but with H-bonds to w1, Lys51, Ser80 and another water molecule (different from w2/w3/w4) (Fig. 3g, Supplementary Fig. 4). The second conformer of Ser48 is rotated ~110° and H-bonded to the N atom of Ala266 and carbonyl oxygen of Leu264. Except for the absence of the w2-w3-w4 water triad, no other structural rearrangements are observed in the vicinity of Ser48. The different hydration state might be the effect of differences in crystal vitrification prior to X-ray data collection. Nevertheless, the structural data indicate that Ser48 is not always hydrated.

Ser48 is H-bonded to Ser80, which is also important for catalysis but even more important for protein stability as it is the kingpin of a network of H-bonds that lead to Lys51 (Nζ), Lys263 (Nζ), Ans134 (Oδ1) and w4. This network stabilizes the position of Ser80 and forces a highly strained, non-planar configuration of the Ala79-Ser80 peptide bond. Replacement of Ser80 or Ser48 by Ala leads to a disruption of the H-bond system in this region and consequently to aberrant main-chain conformation and misfolding.

In the close vicinity of these Ser residues, there are two Lys residues, Lys51 and Lys263. Lys51 is H-bonded to Ser48 and Ser80 and also to Asn134 and Cys135. Lys263 is H-bonded to Ser80, carbonyl oxygen of Leu264 and water w5 (Fig. 3f). Lysine

residues are often involved in catalytic processes, as they can switch their protonation state relatively easily. The positions of Lys51 and Lys263 might suggests that they participate in proton exchange or polarization of Ser80. On the other hand, Lys51 interacts with both Ser80 and Ser48, and also with Cys135 from the coordination sphere of the $Zn^{2+}$ ion (Fig. 3f).

Tracking of the H-bond pattern in the active site revealed an indirect connection between Cys135 from the Zn coordination sphere and a relatively distant Cys249, which seems to carry a chemical modification. The side chain of Cys249 is most likely oxidized and was modeled as S-hydroxycysteine in two conformations (Supplementary Fig. 4). One of the conformers is H-bonded to w6 and Tyr304, while the other is H-bonded via water w5, Lys263, Ser80 and Lys51 to Cys135 (Fig. 3f). The potential origin/role of the Cys249 modification is unclear, but without it the Cys249-w5-Lys263-Ser80-Lys51-Cys135-Zn network cannot be built. In the MS experiments Cys249 was very reactive and was mapped in the MS spectrum as carbamido-methyl cysteine as a result of reaction with iodoacetamide, which is commonly used to prepare MS samples.

**Enzymatic activity of WT ReAV and its active-site mutants.** In qualitative assays of ʟ-asparaginase activity, WT ReAV was most active at alkaline pH 10–12 (Supplementary Fig. 5), in agreement with the pH optimum of 9-11 reported previously[20]. Our quantitative ITC measurements yielded the following kinetic parameters $K_m$, $k_{cat}$ and $k_{cat}/K_m$: $15.8 \pm 1.5$ mM, $278 \pm 25$ s$^{-1}$ and $18 \pm 3$ mM$^{-1}$ s$^{-1}$ at pH 7.5; $6.9 \pm 0.5$ mM, $441 \pm 32$ s$^{-1}$ and $64 \pm 9$ mM$^{-1}$ s$^{-1}$ at pH 8.0; and $4.2 \pm 0.3$ mM, $438 \pm 32$ s$^{-1}$ and $104 \pm 15$ mM$^{-1}$ s$^{-1}$ at pH 9.0 (Fig. 4). Fitting the titration data with the Hill equation showed that the Hill coefficient is different than 1 and equal to $1.4 \pm 0.1$ (at pH 7.5 and 8.0, both with and without $Zn^{2+}$ cations) or to $1.6 \pm 0.1$ (at pH 9.0).

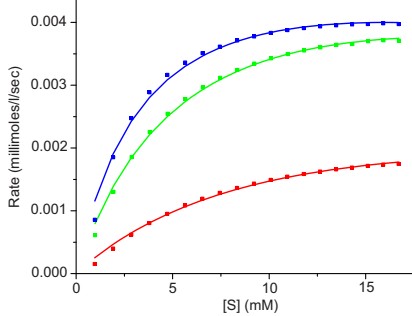

| | $k_{cat}$ [s$^{-1}$] | $K_m$ [mM] | $k_{cat}$ / $K_m$ [mM$^{-1}$·s$^{-1}$] |
|---|---|---|---|
| **pH 9.0** | 438 ± 32 | 4.2 ± 0.3 | 104 ± 15 |
| **pH 8.0** | 441 ± 32 | 6.9 ± 0.5 | 64 ± 9 |
| **pH 7.5** | 278 ± 25 | 15.8 ± 1.5 | 18 ± 3 |

**Fig. 4 Michaelis–Menten analysis of ITC titration data.** Left panel: Michaelis–Menten curves from single heat-rate ITC titrations at pH 7.5 (red), 8.0 (green) and 9.0 (blue). The reaction rates obtained from steady-state heat-flow levels after each injection were plotted against L-Asn concentration and fitted to the Michaelis-Menten equation. Right panel: Summary of the kinetic parameters at each pH condition. The averaged values and their standard deviations were estimated from three separate experiments. Source data are provided as a Source Data file.

The same experiments performed with L-Gln or ampicillin showed no heat effect, indicating that these compounds are not substrates of ReAV. Likewise, spectroscopic measurements did not detect any reaction with the CENTA (a chromogenic cephalosporin) β-lactam antibiotic.

Since the crystal structures revealed that ReAV is a zinc metalloprotein, we determined the kinetic parameters (in 30 mM phosphate buffer pH 8.0) of the enzyme deprived of Zn$^{2+}$ ions (by TPEN treatment) in order to probe the role of Zn$^{2+}$ ions in the L-asparaginase activity. The results are: $K_m$ = 6.5 ± 0.6 mM, $k_{cat}$ = 432 ± 7 s$^{-1}$ and $k_{cat}/K_m$ = 66 ± 7 mM$^{-1}$ s$^{-1}$. We also determined the influence of high concentration of zinc ions by comparing the rate of asparagine conversion without Zn$^{2+}$ and with controlled Zn$^{2+}$ presence at 100 μM, which is about the maximum Zn$^{2+}$ concentration achievable in the bacterial cell[21]. In the case of ReAV deprived of zinc this rate was determined as 1.1 ± 0.1 μM s$^{-1}$ of substrate depletion under the conditions of our experiment. This rate was lowered by 29% in the presence of 100 μM Zn$^{2+}$ ions.

For the five residues implicated by the crystal structures as belonging in the active site area, alanine site directed mutants (S48A, K51A, S80A, C135A and K263A) were generated. Although at room temperature all variants retained the dimeric structure of the WT protein, as indicated by gel filtration (Supplementary Fig. 2), the Nessler test clearly showed that all the mutants were unable to hydrolyze L-Asn (Fig. 5).

**Microcalorimetric measurements of divalent cation binding by ReAV.** Titration of ReAV pre-treated with a strong chelating agent (TPEN), using different divalent metal cations (Zn$^{2+}$, Mg$^{2+}$, Ni$^{2+}$, Mn$^{2+}$, Cd$^{2+}$) revealed that only interaction with Zn$^{2+}$ produces a heat effect (Fig. 6), with the following binding parameters: $N$ = 1.0 ± 0.1, $K_d$ = 2.7 ± 0.9 μM, $\Delta H$ = −4562 ± 200 cal/mol, $\Delta S$ = 10.2 ± 0.6 cal/mol/deg. This indicates a strong and specific binding of the zinc ion by ReAV.

**Structural homologs of ReAV.** The crystal structures of ReAV reveal a unique architecture, with only distant similarity to known protein folds. A DALI[22] search of the PDB found the closest structural homologs among bacterial enzymes (rmsd 1.5–2.2 Å), mostly glutaminases (Fig. 7), e.g., from *Bacillus subtilis* (1mki), *E. coli* (1u60)[23] or *Micrococcus luteus* (3if5)[24] (Supplementary Fig. 6), but also among human or mouse mitochondrial liver-type (4bqm) or kidney-type (4jkt) glutaminases[25]. A more distant similarity (rmsd 2.8–3.1 Å) was detected to various proteins involved in bacterial cell wall biosynthesis or with the ability to bind antibiotics, e.g., penicillin binding protein (PBP) from *E. coli*

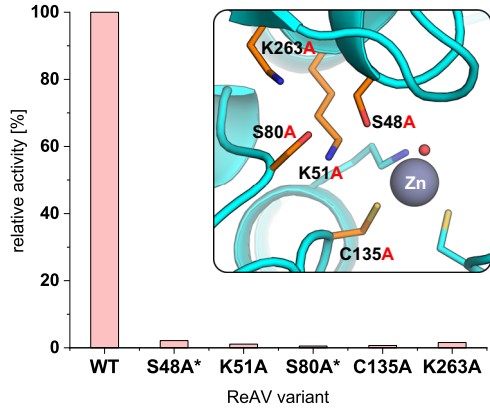

**Fig. 5 L-Asparaginase activity of WT ReAV and its mutants, assessed by the Nessler method at pH 8 in the presence of 10 mM L-Asn.** In each experiment 0.1 mg of the enzyme was used. After 10 min the optical density at 420 nm (OD$_{420}$) was measured. The value of OD$_{420}$ recorded for WT protein marks full activity (100%). Inset: location of the mutation sites in the putative ReAV active site. Mutants S48A and S80A (marked with an asterisk) might have impaired folding, as indicated by CD spectra. Source data are provided as a Source Data file.

(1nzo)[26], *Acinetobacter baumannii* (3ue1)[27], *Staphylococcus aureus* (6c39)[28], *Listeria monocytogenes* (5zqe)[29] or *Neisseria gonorrhoeae* (6hzj, 6p54)[30] (Supplementary Figure 6). Additional homologs (rmsd 2.4–3.2 Å) were found among serine β-lactamases, e.g., from *E. coli* (3qnc, 6skp, 1tem)[31,32], *Klebsiella pneumoniae* (6v1o)[33], *Pseudomonas aeruginosa* (5eua)[34] or *Streptomyces clavuligerus* (2xgn)[35], and peptidoglycan glycosyl-transferase from *Atopobium parvulum* (4r1g, 4jbf)[36] (Supplementary Fig. 6). Sequence similarity between ReAV and the above homologs is low and varies from random to 29% (for *E.coli* glutaminase, 1u60).

## Discussion

Legumes have developed symbiotic relationship with soil bacteria, such as *R. etli*, which fix atmospheric nitrogen in their roots. In this system, the plant supplies the bacteria with carbon sources and L-Glu, while *R. etli* supplies ammonia, L-Asp and L-Ala[37]. In *Rhizobiales*, genes essential for the symbiotic lifestyle are located not only on the chromosome but also on plasmids[38]. In particular, plasmid p42e contains the *asnRPAB* operon[19] encoding proteins involved in the degradation of L-asparagine: L-aspartase (AnsB) and thermolabile (inducible) L-asparaginase ReAV (AnsA)[39]. The gene encoding ReAV might have arisen by horizontal gene transfer from *Eukarya*, especially *Ascomycetes*

fungi[20]. As another curiosity, *R. etli* does not encode any proteins that could be identified with the well-established Classes (1 and 2) of asparaginases found in other organisms.

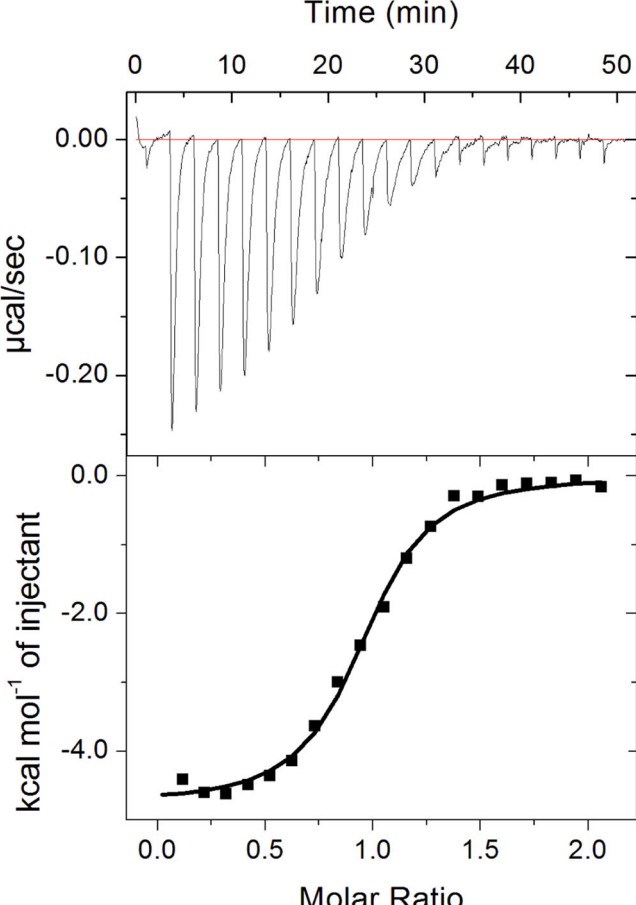

**Fig. 6 Calorimetric titration of ReAV with Zn²⁺ cations.** The top panel shows the raw data obtained from 19 consecutive 2 μL injections of 1 mM ZnCl₂ solution into the sample cell (200 μL) containing 100 μM of the protein. Heat peak areas were plotted against the Zn:protein molar ratio to create the binding isotherm of the bottom panel. The best fit of a model of one set of binding sites is represented by the black line. Source data are provided as a Source Data file.

Our kinetic experiments showed that ReAV hydrolyzes L-Asn with a Michaelis constant of 15.8 ± 1.5 mM and $k_{cat}$ 278 ± 25 s⁻¹ even at pH 7.5, i.e. away from its high pH optimum of 10–12. These results are in agreement within the order of magnitude with the values reported previously for the same assay conditions but using a different method[20] ($K_m$ = 8.9 mM, $k_{cat}$ = 106 s⁻¹). At pH 9.0 the enzyme is much more active, with $K_m$ = 4.2 ± 0.3 mM and $k_{cat}$ = 438 ± 32 s⁻¹. Apart from the quite different Berthelot assay, which differs from the calorimetric method not only in the measured signal (absorbance vs heat) but also in its mechanics (premixed reaction solutions vs fast titration with constant stirring), the previous study used a different protein construct, with a long (7.5 kDa, 20% of protein mass) artificial sequence (His-tag + antibody epitope + protease recognition sequence) attached at the N terminus. However, in terms of enzyme efficiency, the two measurements are very similar ~18 mM⁻¹ s⁻¹ (this study) vs ~12 mM⁻¹ s⁻¹ (Moreno et al.)[20]. To make the two studies comparable, our kinetic curves, with clearly hyperbolic shape (Fig. 4), were fitted to the Michealis-Menten equation. An additional fitting to the Hill equation yielded a Hill coefficient $n_h$ > 1, indicating that ReAV exhibits positive allostery, and this effect was stronger at higher pH ($n_h$ ~1.4 ± 0.1 at pH 7.5 and 8.0 pH vs ~1.6 at pH 9.0). Moreover, zinc-free protein had the same Hill coefficient, indicating that zinc is not the allosteric modulator of ReAV. Examples of allosteric asparaginases belonging to Class 1 were reported before, e.g. L-asparaginase 1 from *Saccharomyces cerevisiae*[40], EcAI and human L-asparaginase I[41]. However, those enzymes exhibited much stronger positive cooperativity and a sigmoidal substrate dependence curve, with Hill coefficients 2.2[40], 3.5 and 3.9[41], respectively. Moreover, they did not follow the Michaelis-Menten kinetics at all. It was established that the substrate itself is the allosteric effector of those enzymes[41].

Our ITC experiments also established that there is no heat effect on interaction with L-Gln, indicating that ReAV is most likely free of L-glutaminase activity. This result is in agreement with earlier observations[20,42]. We couldn't use the simple Nessler test for this reaction because even the highest grade commercial L-Gln (supplied by Sigma-Aldrich/Merck or BioShop) contains impurities that give positive Nessler effect even without any enzyme[43]. L-DON, a typical L-glutaminase inhibitor, did not inhibit the ReAV activity, which indicates that it is not able to bind in the ReAV active site.

All the crystal structures reveal the presence of a zinc ion in the active site area. ReAV affinity for Zn was measured by ITC titration with Zn²⁺ after removal of all metals with a strong chelating agent (TPEN). The results showed that Zn²⁺ binding is

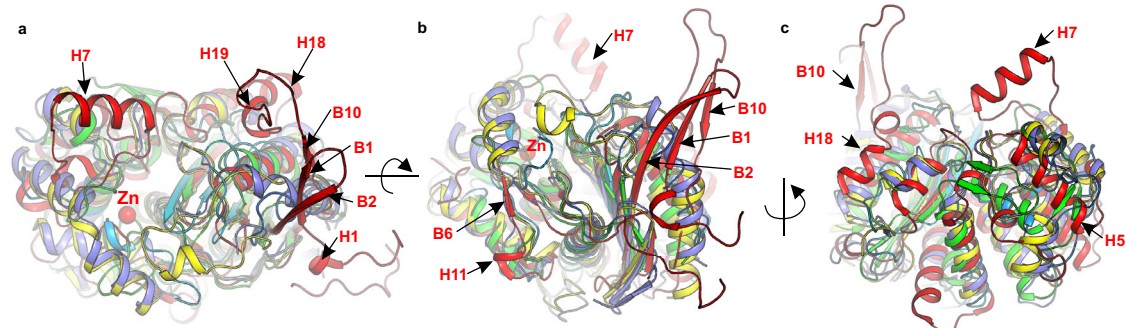

**Fig. 7 Superposition of ReAV (red) and selected proteins with detectable structural similarity (identified by their PDB codes and color).** *E. coli* glutaminase (yellow, 1u60), *B. subtilis* glutaminase (violet; 1mki), human liver-type glutaminase (cyan, 4bqm); *L. monocytogenes* PBP (green, 5zqe). Panels **a**, **b**, **c** show the same superposition in different orientations. The central β-sheet and helices in the protein core superpose quite well, while the most important differences are visible for strands B1, B2, B10 and helix H7. Structural elements that differentiate ReAV from the other proteins are indicated by arrows. The red sphere marks the zinc ion coordinated in the ReAV active site.

strong ($K_d$ ~2.7 μM) and highly selective, as similar divalent cations ($Mg^{2+}$, $Ni^{2+}$, $Mn^{2+}$, $Cd^{2+}$) were not bound by the protein. The kinetic parameters of ReAV treated with TPEN ($K_m$ = 6.5 ± 0.6 mM; $k_{cat}$ = 432 ± 7 s$^{-1}$) are very similar to those of the native protein, suggesting that zinc does not play a catalytic role in L-Asn hydrolysis. Comparison of L-Asn conversion rates by protein devoid of zinc and supplied with maximum physiological concentration of $Zn^{2+}$ (100 μM) in Tris buffer revealed 29% lower activity in the latter case, in general agreement with the results of Moreno et al.[20]. However, we note that Moreno et al. used 1 mM $Zn^{2+}$ in alkaline phosphate buffer, where $Zn_3(PO_4)_2 \cdot 4H_2O$ precipitation occurs[44], and protein construct with a metal-chelating His-tag. Thus, the exact cause of the reported effect was not obvious.

Our enzymatic assays included several ReAV mutants carrying substitutions in the active site area, designed on the basis of our structural analyses. The results clearly show (Fig. 5) that all variants were unable to hydrolyze L-Asn, suggesting that the two serine residues, Ser48 and Ser80, as well as the accompanying Lys51 and Lys263, are crucial for catalysis. Similar lack of activity was observed for the C135A substitution, which would affect the zinc coordination (Fig. 3). As the ITC experiments showed that $Zn^{2+}$ is not necessary for catalysis, it is reasonable to assume that Cys135 and the Zn binding site may be important for protein folding and stability.

The recently proposed classification[1,4] divides L-asparaginases into three classes: bacterial-type (Class 1), plant-type (Class 2) and *R. etli*-type (Class 3)[1,13]. The homology of structure and sequence within the first two classes is high but there is no similarity across the classes. The sequence of ReAV that has been available so far, clearly indicated that *R. etli* defines yet another, completely different class of L-asparaginases, with sequence identity to Class 1 and 2 enzymes at noise level (~15%). In this work we present the structure of the inducible *R. etli* asparaginase ReAV, as the prototypic member of the *R. etli*-type Class 3. ReAV has no structural homology to any known L-asparaginase (or glutaminase-asparaginase). Nevertheless, a UniProt search revealed that a lot of sequence homologs of ReAV (with 35-89% sequence identity) appear in proteobacteria, terrabacteria, and in filamentous ascomycetes fungi.

The structural homologs of ReAV found in the PDB by DALI belong to different protein families, viz. glutaminases, PBPs and β-lactamases. Superposition of ReAV on these PDB models revealed relatively high structural similarity in the α-helical domain for helices H2, H4, H9, H10, H12, H13 and H17, with most significant differences seen in the region of helix H1 (and the N terminus), H7, H18 and H19 (Fig. 7). A degree of similarity is also visible at the β-strands B2, B3, B8, B7 and B9; however, in the superposed homologs the β-strand corresponding to B2 is shorter and strands B1 and B10 are absent altogether (Fig. 7). Some structural homologs of ReAV have evolved additional domains, thus a degree of structure similarity is visible only for the catalytic domain but not for the peripheral motifs that are absent in ReAV (Supplementary Fig. 6).

Interestingly, the ReAV dimerization motif formed by the β-strands B2, B1, B10, is replaced in many of the analyzed glutaminases, β-lactamases and PBPs by a long α-helix (Supplementary Fig. 7). However, this α-helix is not involved in dimer formation of these proteins. Not all proteins listed above are dimeric. The β-lactamases from *E. coli*[31] and *K. pneumoniae*[45] are dimeric, but they dimerize utilizing short antiparallel β-strands originating from the protein core (Supplementary Fig. 7). The elements corresponding to helix H7 and strands B1, B2 and B10, that are responsible for ReAV dimerization, are absent in those β-lactamases.

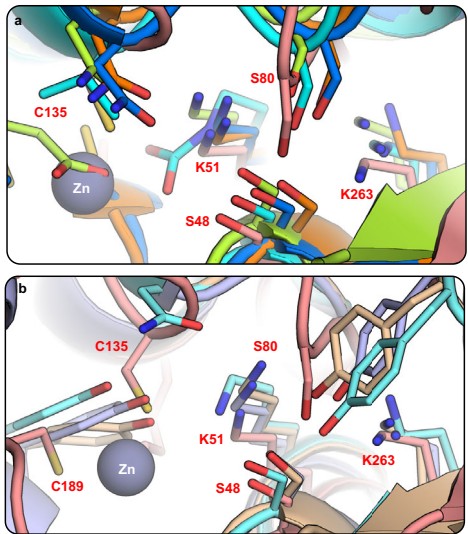

**Fig. 8 Superposition of the catalytic centers of ReAV (salmon, red residue numbers) and selected structural homologs (identified by their PDB codes and color). a** β-Lactamases from *K. pneumoniae* (cyan, 6v1o, the counterpart of K51 is acetylated) and *P. aeruginosa* (light green, 5eua); peptidoglycan glycosyltransferase from *A. parvulum* (blue, 4r1g); PBP form *N. gonorrhoeae* (orange, 6hzj). All these proteins possess two Ser-Lys tandems in the active site. **b** In some structural homologs of ReAV, Ser80 is replaced by Tyr, e.g., glutaminase from *B. subtilis* (violet, 1mki), human liver-type glutaminase (wheat, 4bqm) and kidney-type mouse glutaminase (light cyan, 4jkt). The gray sphere marks the Zn cation in the ReAV structure.

These observations suggests that ReAV might have common evolutionary ancestors with bacterial glutaminases, lactamases or PBPs. This conjecture is supported by the observation that the putative active site of ReAV corresponds quite well to the active sites of these structural homologs, albeit only in the part encompassing Ser48, Ser80, Lys51 and Lys263 (Fig. 8). In most of the structural homologs, the active site is located in a region of the protein core corresponding to the ends of helices H2, H4, H9, H13 and β-strand B7 (Fig. 3c). In these structures, the catalytic Ser maps to Ser48 of ReAV and the counterparts of Ser80, Lys51 and Lys263 are also preserved. However, in some of the homologs the hydroxyl group corresponding to Ser80 is provided by a Tyr residue (Fig. 8).

These observations strongly suggest that the catalytic nucleophile of ReAV is Ser48. In most of the structures presented in this study, Ser48 is tightly surrounded by three water molecules, w2-w3-w4, in a nearly tetrahedral arrangement and with excellent definition in the electron density maps (Fig. 3e). A similar triad was also found in the crystal structure of *B. subtilis* glutaminase (1mki). However, it was interpreted in that structure as a phosphate group covalently linked to the putative nucleophilic serine, despite the fact that MS experiments performed by the authors did not confirm phosphorylation[23]. Likewise, MS experiments carried out for ReAV did not reveal any phosphorylation, or any covalent modification of Ser48 for that matter. Consequently, the three electron density peaks surrounding the Oγ atom of Ser48 in ReAV are interpreted as water molecules.

The presence in ReAV of two serine-lysine dyads, Ser48-Lys51 and Ser80-Lys263, resembles the arrangement of the catalytic residues in the model β-lactamases and other ReAV homologs (Fig. 8). Their catalytic mechanism was studied in detail[46], revealing that the counterpart of Ser48 acts as the nucleophile, while the assisting lysine (Lys51 in ReAV) polarizes the hydroxyl

group of the nucleophile, and also activates a water molecule in the second step of the hydrolytic reaction. The second Ser-Lys pair, corresponding to Ser80-Lys263 of ReAV, is involved in substrate binding. It was concluded that for its role in catalysis, the counterpart of Lys51 must be unprotonated, i.e., in free-base form[47]. It was also found that both lysines contribute to the high pH optimum (9.5–10.5) for PBP5[47] catalysis. This observation is in excellent agreement with our findings that ReAV is most active at alkaline pH.

We can conclude that in ReAV all four residues, Ser48, Lys51, Ser80 and Lys263, are also important for L-Asn hydrolysis, as their mutations rendered the enzyme inactive. In this view of the reaction, Ser48 launches a nucleophilic attack on the amide group of the substrate, leading to an acyl-enzyme intermediate and $NH_3$ release. In the second step of the reaction, the ester intermediate is hydrolyzed by an activated water molecule. For the first step of the reaction, the Ser48 nucleophile is strongly polarized by Lys51. The attraction of the three water molecules w2-w3-w4 by the Ser48 side chain is a clear evidence of this polarization. In the absence of a substrate, the arrangement of w2-w3-w4 might mimic the tetrahedral transition state of the catalytic event (Supplementary Fig. 8). However, in our numerous collection of ReAV structures, we also see cases with incomplete or absent (MP2) water triad. The corresponding crystals were grown at different pH, detergent and salt conditions, which would certainly affect the protonation state of the key lysine residues and, therefore, polarization of Ser48. It can be hypothesized that in the crystal structure with no w2-w3-w4 triad and with double conformation of Ser48 (MP2), the hydroxyl group of Ser48 is not polarized and this structure shows the enzyme in its resting state, between catalytic cycles.

ReAV has yet another characteristic residue, namely Cys249 with an oxidative modification at the Sγ atom, located ~5 Å from the putative nucleophilic Ser48. The physiological function of this cysteine is unclear; however, symbiotic *Rhizobium* species have a very intensive metabolism, generating high amounts of reactive oxygen species (ROS)[48]. Oxidative modifications of cysteine side chains play an important protective role against ROS[49,50]. In ReAV, Cys249 may be a "suicide" quencher, protecting the sensitive zinc binding site from oxidative damage. Cys249 is not only very reactive, but is involved in a network of H-bonds that extends to the Ser-Lys dyads and to Cys135 in the Zn coordination sphere. Cys249 does not have a counterpart in the structural homologs of ReAV.

Except ReAV, there are no other L-asparaginases with a metal ion right in the catalytic center. Previously, zinc binding sites were also found in the *E. coli* enzyme EcAII, however, the metal ions were located distinctly away from the active site[51]. It must be emphasized that none of the analyzed structural homologs of ReAV contains a metal ion near the active site, close to the Ser-Lys pairs. The presence of $Zn^{2+}$ very close to the two Ser-Lys pairs, and coordinated by an additional Lys residue, is a highly unusual feature, uniquely characteristic of ReAV. Interestingly, in the structurally homologous PBPs, lactamases and glutaminases, the space occupied by the $Zn^{2+}$ cation in ReAV is preserved, but is usually filled with a Tyr or Glu side chain (Fig. 8) that is often involved in catalysis.

The $Zn^{2+}$ ion in the active site of ReAV has an unusual coordination sphere formed by two cysteines, a lysine, and a water molecule (Fig. 3). Our ITC experiments revealed that Zn binding is highly selective and specific. On the other hand, our kinetic data show that zinc is not necessary for L-Asn hydrolysis. The enzyme kinetic data contradict, therefore, the conclusion about the role of the $Zn^{2+}$ cation that might be drawn from its open coordination sphere. We cannot exclude, of course, that ReAV might also have other catalytic properties and act as an

enzyme with dual (or even multiple) activity and that for the other activity zinc might be necessary. This speculation, however, would need a lot of experimental work (and even serendipity) to be verified. To date, we have only checked the potential β-lactamase and glutaminase activities and received negative results.

Finally, *R. etli* is also interesting as an evolutionary curiosity. How did it lose the "standard" asparaginases found in other bacteria? And how did it acquire asparaginase activity in an entirely unrelated protein? Is it possible that a serine β-lactamase (not to be confused with zinc metallo-β-lactamase) was hijacked as a scaffold for an asparaginase in *R. etli* (or its ancestor) by blocking the original activity by Zn coordination? In this speculative scenario Zn acts as an inhibitor by sterically blocking the original active site. Is ReAV a freak of nature or can similar enzymes be found in other groups of organisms? The questions can go on as this is only the beginning of a fascinating research project.

## Methods

**Cloning, mutagenesis, expression, and purification**. The gene encoding inducible *R. etli* (strain CFN 42) asparaginase (wild type, WT) was obtained from Deutsche Sammlung von Mikroorganismen und Zellkulturen GmbH, and cloned to pET151-D-TOPO expression vector (Invitrogen) carrying a TEV-cleavable N-terminal His6-tag. Protein production in the *E. coli* strain *BL21 STAR (DE3)* (Invitrogen) was induced with 0.5 mM IPTG. Cells were grown overnight at 16 °C and the expression level was checked by SDS-PAGE. Harvested cells were resuspended in 100 mM Tris buffer pH 8.0, containing 500 mM NaCl and 10 mM imidazole. Cells were disrupted by sonication and centrifuged cell lysate was loaded on a chromatography column filled with HisTrap Ni resin (GE Healthcare). ReAV was eluted form the column using imidazole gradient (to 500 mM). Fractions containing ReAV were pooled and dialyzed overnight in the presence of TEV protease at 4 °C against 100 mM Tris buffer pH 8.0, containing 100 mM NaCl and 5 mM TCEP. His-tag debris (including His-tagged TEV protease) was removed by another HisTrap Ni column. Finally, the protein was loaded on HiLoad 16/600 Superdex 200 (GE Healthcare) gel filtration column and eluted using 100 mM Tris pH 8.0 with 100 mM NaCl.

Protein purity and homogeneity was checked by SDS-PAGE and DLS. L-Asparaginase activity was routinely monitored colorimetrically at 420 nm by the Nessler method[52] using 10 mM L-Asn (in 20 mM Tris buffer pH 8.0) as substrate and commercially available $K_2[HgI_4]$ (Chempur). The same method was used to find the optimum pH of the enzymatic reaction, but with Tris buffer replaced by a series of 25 mM Britton-Robinson buffers covering pH 2-13. The pH activity profile was additionally confirmed by Berthelot reaction[53].

Protein identity was confirmed by MS as follows. Protein was reduced with $NH_4HCO_3$ and DTT at 95 °C; alkylation was carried out with iodoacetamide. Protein was digested overnight at 37 °C with sequencing-grade trypsin (Promega). Peptide solutions were deposited on a MALDI AnchorChip plate (Bruker Daltonics GmbH) with saturated solution of α-cyano-4-hydroxycinnamic acid in 50% ACN and 0.1% TFA (matrix). Spectra were acquired with a minimum of 2500 laser shots in a MALDI TOF/TOF UltrafleXtreme mass spectrometer (Bruker Daltonics GmbH). The MS/MS data of combined peptide mass fingerprints were used for searching UniProt database installed on Mascot server (Matrix Science, London, UK).

Site-directed mutagenesis of ReAV was carried out according to Q5-site mutagenesis protocol (New England Biolabs) using site-specific non-overlapping mutagenic primers (Supplementary Table 1) to introduce substitutions into the following positions: S48A, S80A, K51A, K263A, and C135A. Presence of mutations was confirmed by sequencing (Genomed S.A., Poland). Mutants were produced and purified as described for WT protein. Their activity was checked by the Nessler method using the same protocol as for WT ReAV.

**Folding, monitoring of oligomeric state and thermal stability**. The correct folding and thermal stability were routinely checked for the WT and mutant proteins using circular dichroism (CD) and nano-scale differential scanning fluorimetry (nanoDSF). Prior to the measurements, protein samples were applied to analytical Superdex75 10/300 GL column (GE Healthcare) and were eluted using 50 mM phosphate buffer pH 7.5 with 20 mM NaCl. These experiments were also used to assess the oligomeric state of the proteins. Far-UV CD spectra were recorded using a Jasco J-815 spectropolarimeter. Protein concentration was kept in the 5–13 μM range. Three scanning acquisitions were accumulated and averaged to yield the final spectrum.

Thermal stability of WT and mutant ReAV proteins was assessed by nanoDSF using Prometheus NT.48 (NanoTemper Technologies) and sample volume of ~10 μl per capillary. Fluorescent emissions at 330 and 350 nm showed that all samples were within the detectable concentration range, however, the lowest fluorescence was observed for the S48A and S80A mutants. Melting scans were

recorded as fluorescence emission for protein samples subjected to a temperature ramp of 60 °C/hr from 20 to 95 °C.

Detailed WT ReAV melting profile at different pH and ionic strength was monitored by differential scanning calorimetry (DSC) using a MicroCal PEAQ-DSC system (Malvern Instruments Ltd.). Each DSC experiment consisted of two measurements, both at the same conditions: (i) five reference scans with buffer-filled cells to establish instrument thermal history and to achieve near perfect baseline repeatability; (ii) one sample-buffer scan to acquire data for analysis. The best experiment conditions were: temperature range 25–70 °C, scan rate 15 °C/hr, and low feedback mode. Protein concentration varied between 100 and 500 μM. Before the DSC experiments protein was dialyzed against specified buffers: 50 mM Tris of pH 6.5, 7.5, 8.5, or 50 mM Tris pH 7.5 containing 10 mM NaCl. All buffers contained 3 mM TCEP.

**Microcalorimetric assessment of divalent metal cation binding.** In order to remove any bound $Zn^{2+}$, prior to ITC titrations the WT protein was incubated for 2 h on ice with 2-fold excess of TPEN (N,N,N',N'-tetrakis(2-pyridinylmethyl)−1,2-ethanediamine) dissolved in 96% ethanol. After incubation, TPEN was removed by dialysis and gel filtration on a HiLoad 16/600 Superdex 200 (GE Healthcare) column (50 mM Tris pH 8.0, 100 mM NaCl, 1 mM TCEP).

Titrations were performed at 25 °C using a MicroCal iTC200 calorimeter (GE Healthcare). The protein in the cell at ~100 μM concentration (determined at 280 nm) was titrated with 2 μl aliquots of 1 mM $XCl_2$ salts (X = Zn, Mg, Ni, Mn, Cd). Both, metal and protein were in 25 mM Tris buffer pH 8.0, with 100 mM NaCl and 1 mM TCEP. The raw ITC data were analyzed with the Origin 7.0 software (OriginLab) to obtain the thermodynamic parameters of the complexation reactions: stoichiometry (N), dissociation constant ($K_d$), as well as changes in enthalpy (ΔH) and entropy (ΔS). The measurements were performed in duplicate.

**Measurements of enzymatic activity of ReAV.** Kinetic parameters of the L-asparaginase reaction of ReAV were measured using ITC in the multiple injection mode (MIM)[54] and MicroCal iTC 200 (GE Healthcare) and MicroCal PEAQ-ITC (Malvern) calorimeters. The measurements were taken in 30 mM phosphate buffer of pH 7.5, 8.0 and 9.0. The MIM method consists of two separate experiments. First, the enthalpy of total conversion of all the substrate into product (total molar enthalpy of the reaction, $H_{app}$) was determined by injecting 2 μl of 10 mM substrate (L-asparagine or L-glutamine or ampicillin) (Sigma–Aldrich) into the reaction cell containing 2 μM enzyme. Three injections were performed, separated by intervals long enough to ensure total substrate conversion. After integration of each peak, the enthalpies of all injections appeared to be comparable and were averaged to obtain $H_{app}$. Next, the differential power change (dQ/dt) arising from the turnover of the substrate into product was determined in a heat-rate shift experiment, in which the substrate at 100 mM concentration (in the syringe) was injected in nineteen 2 μL aliquots with short 60 s intervals (to minimize substrate depletion) into the cell with the enzyme kept at ~14.7 nM concentration. All measurements were taken at 37 °C, with stirring at 700 rpm and differential power set to 5 μcal/s. The raw rate data were analyzed using the Enzyme Assay Method 2 included in the ITC DataAnalysis option of the Origin 7.0 software (OriginLab). Briefly, they were transformed into reaction rates and L-asparagine concentrations and fitted to Michaelis-Menten equation. Final kinetic parameters at pH 7.5, 8.0 and 9.0 were calculated by averaging the values obtained in three separate experiments. The data were additionally fitted to the Hill equation (allosteric sigmoidal graph built in GraphPad Prism 6 for Windows) to verify the allosteric effect of the enzyme.

For testing the hypothetical β-lactamase activity of ReAV, an additional assay was applied, utilizing the convenient spectroscopic properties of the yellow chromogenic CENTA β-lactam substrate. Briefly, the reaction was monitored at 22 °C (at which the lactam is stable) for 10 min at 340 nm (absorption maximum of CENTA) and 405 nm (absorption maximum of the product of the reaction) in 1 mL cuvette containing 1 mM CENTA and 2 μM of ReAV in the buffer used in microcalorimetric experiments.

The impact of $Zn^{2+}$ on ReAV activity was assessed by comparing the rate of asparagine (at 10.1 mM concentration in 222 μL volume) conversion by 13.2 nM protein, deprived of zinc by the TPEN procedure or supplied with 100 μM of $ZnCl_2$. The ITC MIM measurements were taken in 5 mM Tris buffer pH 8.0 with 10 mM NaCl to avoid precipitation of insoluble zinc phosphate. In addition, full kinetic characterization of TPEN-treated ReAV was carried out in 30 mM phosphate buffer pH 8.0.

**Crystallization.** Prior to crystallization, WT ReAV was dialyzed to 50 mM Tris buffer pH 8.0 containing 3 mM TCEP and concentrated to 20–25 mg/ml using Amicon 10 kDa spin columns (Merck-Millipore). Crystallization conditions were systematically screened using detergent and additive screens (Hampton Research). Crystals were grown at 19 °C using vapor diffusion in the hanging drop setup. The best single crystals, representing several crystal forms, were obtained in the following conditions: 25% PEG3350, 0.2 M $Li_2SO_4$, 0.01% (w/v) heptane-1,2,3-triol, 0.1 M Tris pH 8.0 (orthorhombic $P2_12_12_1$, OP, START); 25% PEG3350, 0.2 M $Li_2SO_4$, 0.1 M Tris pH 8.0, ~20 mM n-octyl-β-D-thioglucoside (monoclinic $P2_1$, MP1); 2% PEG8000, 20% PEG400, 5 mM $Mg(CH_3COO)_2$ pH 6.5, ~6 mM sucrose

monolaurate (monoclinic $P2_1$, MP2); and 2% PEG8000, 20% PEG400, 5 mM $Mg(CH_3COO)_2$ pH 6.5, ~0.5 mM CYMAL-3 (monoclinic C2, MC). Before data collection, crystals were cryoprotected in their mother liquor supplemented with ethylene glycol or glycerol and vitrified in liquid nitrogen.

**X-Ray data collection, structure solution and refinement. Structural analysis.** X-Ray diffraction data were collected using synchrotron radiation from beamlines at Petra III EMBL/DESY in Hamburg or BESSY[55] in Berlin. Initially, the diffraction data were collected at EMBL beamline P13[56] at the Petra III storage ring. Since in the 367-residue amino acid sequence of ReAV there are as many as 21 Cys/Met residues, long-wavelength (λ = 2.0664 Å, 6.0 keV) anomalous X-ray diffraction data were collected for S-SAD phasing. The data were measured at seven positions of a rod-shaped crystal with approximate dimensions 350 × 200 × 150 μm. At each position, 3600 frames of 0.1° oscillation and 40 ms exposure were collected. The anomalous data (as well as all subsequent X-ray diffraction images) were indexed and integrated with XDS and further processed with XSCALE[57]. Out of the seven anomalous data sets, the best four were selected, scaled and merged together to serve as an experimental S-SAD data set (START). A high-resolution native data set (1.29 Å, OP) was collected at λ = 0.9762 Å for the same crystal.

The data were input to SHELXC and SHELXD[58] for sulfur-substructure determination using the HKL2MAP interface[59]. An iterative density modification and main-chain auto-building as implemented in SHELXE was used to build the initial map. ARP/wARP[60] was used to build the first model (START), with ~95% docked residues and with the initial refinement R value of ~35%. The model was manually rebuilt with Coot[61] and refined using REFMAC5[62]. After structure solution, a high electron density peak was found close to Ser48, suggesting coordination of a metal cation. The peak height and tetrahedral coordination were consistent with $Zn^{2+}$. The metal identity was confirmed by an X-ray fluorescence scan in the range of Zn absorption edge (Supplementary Fig. 9). Since no zinc was used in the crystallization conditions, the metal must have been bound by the protein during expression/purification.

The consecutive structures discussed here were solved by molecular replacement using Phaser[63]. Structures were refined with REFMAC5 using anisotropic (OP) or TLS (START, MP1, MP2, MC) protocols. The electron density maps were inspected in Coot. The statistics of data collection and structure refinement are summarized in Supplementary Table 2.

The structures were analyzed and visualized using PyMOL[64], DALI[22] and PISA[65]. The electrostatic potential distribution was calculated using Adaptive Poisson-Boltzmann Solver (APBS) algorithm[66–68] as implemented in the PyMOL APBS Plugin. Due to low sequence similarity, superpositions of ReAV and its structural homologs identified by DALI were calculated using the Secondary Structure Matching tool[69] from the ccp4 package[70]. Sequence similarity between ReAV and selected structural homologs was assessed by pairwise sequence alignments in EMBOSS Needle[71]. Additional software used in this work is listed in Supplementary References.

**Reporting summary.** Further information on research design is available in the Nature Research Reporting Summary linked to this article.

## Data availability
Atomic coordinates and structure factors corresponding to the final crystallographic models of ReAV generated in this study have been deposited in the Protein Data Bank (PDB) under accession codes 7os3 (START), 7os5 (OP), 7os6 (MP1), 7ou1 (MP2), and 7oz6 (MC). The corresponding raw diffraction images have been deposited in the Macromolecular Xtallography Raw Data Repository (https://mxrdr.icm.edu.pl) under DOI numbers: 10.18150/74YTYQ (START), 10.18150/MUKYJI (OP), 10.18150/XBX6JE (MP1), 10.18150/VQQIHQ (MP2), and 10.18150/6W9HGI (MC). Source data are provided with this paper.

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

## Acknowledgements

We thank Dr. Milosz Ruszkowski for crystal vitrification and participation in synchrotron data collection at EMBL/DESY and Dr. Jakub Barciszewski for data collection at BESSY. We thank Dr. Piotr Bonarek and Dr. Jakub Nowak for help with the CD and nanoDSF experiments. We acknowledge Dr. Lukasz Marczak for carrying out the MS measurements. We thank the Helmholtz-Zentrum Berlin für Materialien und Energie for the allocation of synchrotron beamtime, and Dr. Thomas Schneider for his interest in the S-SAD experiment at EMBL/DESY. Work supported by National Science Centre (NCN, Poland) grant 2020/37/B/NZ1/03250.

## Author contributions

J.I.L. designed the experiments, performed site-directed mutagenesis, grew crystals, refined the structures, analyzed the results and wrote the manuscript. B.I. produced and purified the protein, grew crystals, refined the structures, analyzed the results and participated in manuscript preparation. J.S. carried out enzymatic assays, analyzed their results and participated in manuscript preparation. A.W. carried out expression of ReAV mutants and biophysical measurements. M.B. established the preliminary crystallization conditions and conducted the DSC experiments. M.G. collected and processed X-ray diffraction data, solved the structures, analyzed the results and participated in manuscript preparation. M.J. conceived and coordinated the project, designed the experiments, analyzed the results and wrote the manuscript.

## Competing interests

The authors declare no competing interests.
