## [Peer Review File · Nature Communications]

Crystal structures of the elusive *Rhizobium etli* L-asparaginase reveal a peculiar active siteREVIEWER COMMENTS

Reviewer #1 (Remarks to the Author):

The manuscript of Prof. Jaskolski et al. represents an important contribution in the field, since as the title clearly indicates, the atypical asparaginase from *Rhizobium etli*, had remained uncharacterized a long time ago. Now, thanks to this study, it is experimentally verified that the active site of asparaginase and its 3-dimensional conformation are structurally different from the known asparaginases used in ALL treatment.

I found the article very interesting, clear, and concise. It contains crystallographic data of asparaginases in their native state and with point mutations, which allows a very in-depth analysis of their structure and catalytic site. I think that it may be ready for publication after some clarification that I describe below:

1. The asparaginases currently used in the ALL treatment have a substrate affinity in the micromolar range, a necessary condition for their functioning under physiological conditions. Previous studies (Moreno-Enriquez, et al., 2012) showed that the *R. etli* asparaginase activity has a K_m in the millimolar range. How is this discrepancy explained? At what pH the K_m for asparagine was determined? Should it be determined at physiological pH and not at the one in which there is the maximum enzymatic activity? Please clarify.
2. An interesting finding is the presence of Zn interacting with catalytic residues. In this work, it is shown that Zn and no other divalent cations interact with the enzyme. In previous works, it was reported that, in addition to Zinc, the presence of Manganese affects the enzymatic activity. Can you explain this discrepancy in results?
3. Since *R. etli* asparaginase has a catalytic activity with the participation of uncommon residues absent in other asparaginases, it would be very illustrative to add a figure exemplifying the proposed hydrolytic deamidation of asparagine in the active site and the role of catalytic residues.
4. How do they explain that some of the point mutations of residues that are part of the catalytic site (S48, S80), resulted in an important alteration in the protein folding, which is reflected in its loss of solubility and incorporation into inclusion bodies?
5. Some minor changes:

Page 6, Line 149: Ligands

Page 8, line 215: Michaelis-Menten

Page 8, line 222: Please include "CENTA" definition

Reviewer #2 (Remarks to the Author):

In their manuscript "The elusive *Rhizobium etli* L-asparaginase ... ", Loch et al. describe the structures, physicochemical characteristics and kinetic properties of asparaginase from *R. etli* (ReAV), native and its mutant forms. The authors employed quite a wide range of techniques in their studies, with X-ray crystallography being the dominant one. Although ReAV was described and to some extent characterized many years ago (Moreno-Enriquez et al., *J. Microbol. Biotechn.*, 2012, 22, 292-300, furthermore ME2012), this is the first report characterizing this enzyme structurally. As far as structural characterization is concerned, the presented material is of very high quality, in fact my overall impression is that it dominates the message of this report and as such would be more suitable for a journal focused on structural properties. Whereas this element can easily be corrected, I have a number of concerns regarding interpretation of results presented here.

- 1) In several instances the authors inform us about potential of ReAV for future utilization in the treatment of a human disease (leukemia). Unfortunately, I can't see this possibility. First, catalytic properties of ReAV are completely incompatible with those expected from anti-leukemia therapeutics. Values of K_M and k_{cat} (0.5 mM and 1.8 s⁻¹) are far from the desired values and it is extremely unlikely that this enzyme can be "optimized" to acceptable target values. Even these authors report

that "optimal" activity of ReAV is observed at pH range 10-12 (very different from physiological conditions in human). Furthermore, similar data reported earlier for ReAV were vastly different (K_M and k_{cat} were 8.9 mM and 108 s⁻¹) (ME2012) and those authors reported optimal activity at 50°C (a potential problem in terms of therapeutic use). Taking into account that such fundamental characteristics as protease resistance (i.e. plasma half-life) and immunogenic reactivity are not evaluated here at all, a suggestion about a potential therapeutic use is a stretch. Additionally, the authors of this report don't discuss at all the great discrepancy between the results of their kinetic studies and those published earlier.

2) Glutaminase activity. I am a bit surprised that the authors did not report results from a simple assay using the Nessler reagent, monitoring progress of a reaction between ReAV and L-Gln, instead of employing ITC technique. Lack of the thermal effect during mixing (and a possible reaction) of ReAV and L-Gln does not indicate at all the absence of catalytic reaction. It may simply suggest that the reaction is primarily associated with entropic effects. In such a not uncommon case ITC is not the appropriate technique to study this reaction. It is quite possible that ReAV is devoid of L-glutaminase activity, but the results presented here are not demonstrating it conclusively.

3) Appearance (identification) of Zn²⁺ cation is discussed in this report quite extensively, however, I would expect to see some comments on the differences between the reported results and those described earlier (ME2012). Also, another potential problem involves the determination of the identity of the Zn²⁺. Although ITC studies seem quite compelling (in terms of associated ΔH), how do the authors reconcile their results with those published earlier (ME2012), in which several divalent cations (including Zn²⁺) were found inhibitory for catalysis. Probably, much stronger support for identification of the nature of the cation could be achieved by anomalous diffraction experiments?

Additional comments:

a: on few occasions the authors refer to "current classification of asparaginases" with the reference to their 2021 paper. I think that it is much more appropriate to refer to "recently proposed classification of asparaginases" as it is not yet universally accepted and certainly not "generally current classification"

b: the reference 16 on page 4 (line 82) is incorrect

c: the authors acknowledge that the structure of ReAV is similar to some glutaminases. The rmsd value between ReAV and these enzymes, close to 1.2 Å, suggests very close similarity. Therefore, their claim of novelty of the ReAV structure needs to be considered much more carefully. Asparaginase and glutaminase activities are chemically closely-related and more compelling arguments need to be presented to claim the uniqueness of ReAV in this area.

In summary, this report represents a high-quality structural description of ReAV with a series of physicochemical and kinetic experiments characterizing this enzyme. Whereas I have an impression that structural details play a dominant role, I am not convinced about the interpretation of some physicochemical and kinetic data. Additionally, I do not believe in the postulated link between the results presented here and a potential therapeutic usefulness of this enzyme.

NCOMMS-21-25168 – Responses

Reviewer #1 (Remarks to the Author):

The manuscript of Prof. Jaskolski et al. represents an important contribution in the field, since as the title clearly indicates, the atypical asparaginase from *Rhizobium etli*, had remained uncharacterized a long time ago. Now, thanks to this study, it is experimentally verified that the active site of asparaginase and its 3-dimensional conformation are structurally different from the known asparaginases used in ALL treatment.

I found the article very interesting, clear, and concise. It contains crystallographic data of asparaginases in their native state and with point mutations, which allows a very in-depth analysis of their structure and catalytic site.

We thank the Reviewer for this positive opinion about our work.

I think that it may be ready for publication after some clarification that I describe below:

1. The asparaginases currently used in the ALL treatment have a substrate affinity in the micromolar range, a necessary condition for their functioning under physiological conditions. Previous studies (Moreno-Enriquez, et al., 2012) showed that the *R. etli* asparaginase activity has a K_m in the millimolar range. How is this discrepancy explained? At what pH the K_m for asparagine was determined? Should it be determined at physiological pH and not at the one in which there is the maximum enzymatic activity? Please clarify.

We have repeated the kinetic measurements using the same conditions (phosphate buffer, pH 7.5, enzyme and substrate concentrations) as Moreno-Enriquez, et al. 2012 (ME2012) and received similar results ($K_m=15.8\pm 1.5$ mM) as ME2012 ($K_m=8.9$ mM). The difference may be attributed in part to the different method used (in ITC the reaction solution is constantly stirred, improving substrate-enzyme diffusion), and in part to the rather unnatural protein construct, containing a 7.5-kDa (20% of WT protein mass) N-terminal attachment (His-tag + antibody epitope + protease recognition sequence) used in the ME2012 study. In contrast, our protein had the His-tag removed. However, the enzyme efficiency from the two studies at pH 7.5 is very similar: ~ 18 $\text{mM}^{-1}\cdot\text{s}^{-1}$ (our study) and ~ 12 $\text{mM}^{-1}\cdot\text{s}^{-1}$ (ME2012). In the revised manuscript, we present the newly determined kinetic parameters in comparison with the ME2012 results, and discuss the possible source of differences. The above measurements were taken at physiological pH (7.5). For comparison, we have also determined the kinetic parameters at pH 8.0 and 9.0, to confirm the increase of enzymatic efficiency in alkaline conditions. In the revised manuscript, we clearly state the pH conditions of the reported kinetic parameters. The results of the new ITC experiments conducted at pH 7.5, 8.0 and 9.0 are presented as new Fig. 4.

We feel we need to explain the source of the difference in the kinetic parameters established in the original and revised manuscripts. We were able to track down the source of the misestimated parameters presented in the original submission to insufficient substrate (relative to enzyme) concentration. While the measurements as such were correct, they were taken in a reaction mixture with too low L-Asn concentration (due to solubility problems we had at that time), which did not allow the enzyme to reach V_{max} . We are very grateful to both Reviewers for the motivation to repeat the experiments and correct these parameters.

2. An interesting finding is the presence of Zn interacting with catalytic residues. In this work, it is shown that Zn and no other divalent cations interact with the enzyme. In previous works, it was reported that, in addition to Zinc, the presence of Manganese affects the enzymatic activity. Can you explain this discrepancy in results?

The previous authors were not aware of the fact that ReAV is a Zn-binding protein. Since prior to other Me^{2+} binding studies the natural Zn^{2+} cation had not been removed, it is difficult to explain the previous results because the interference of Zn^{2+} with the additional cations used in the test reactions is not known. Moreover, as noted above, the protein used by ME2012 contained the metal-binding His-tag, which is very likely to interfere with any experiments that use divalent metal cations. Also, we note that the Zn^{2+} (and other metal ion) concentration of 1 mM used by ME2012 was highly non-physiological. Above all, however, ME2012 conducted their metal-binding studies in phosphate buffer at alkaline pH, where Zn^{2+} (and many other divalent cations) are precipitated as insoluble phosphates (e.g. $\text{Zn}_3(\text{PO}_4)_2 \cdot 4\text{H}_2\text{O}$). Those results cannot be treated, therefore, with confidence. A comment about the shortcomings of the previous results has been included in the revised manuscript. Nevertheless, we have conducted additional experiments to shed more light on the influence of high levels of zinc ions on the asparaginase activity of ReAV. Of necessity, we had to use a different buffer (Tris instead of phosphate) but the pH was as in ME2012 (8.0). We show that at the maximal physiological concentration of Zn^{2+} cations (100 μM), which is more than enough to oversaturate the protein, the activity is reduced by 29%. We know that this effect is non-specific because in precise titration the protein: Zn^{2+} stoichiometry is 1. As described in the manuscript, our ITC experiment did not show any specific binding of Mn^{2+} cations by ReAV.

3. Since *R. etli* asparaginase has a catalytic activity with the participation of uncommon residues absent in other asparaginases, it would be very illustrative to add a figure exemplifying the proposed hydrolytic deamidation of asparagine in the active site and the role of catalytic residues.

In view of the absence of a crystal structure of an ReAV in complex with a substrate/product, a detailed mechanism would be highly speculative. What we can (and do) say with reasonable confidence is that S48, activated by K51, would act as the primary nucleophile attacking the C_γ atom of the side-chain amide group of the substrate, leading to an acyl-enzyme intermediate that would be hydrolysed in the second step of the reaction by a water molecule. A paragraph about this hypothetical mechanism has been included in the revised manuscript.

4. How do they explain that some of the point mutations of residues that are part of the catalytic site (S48, S80), resulted in an important alteration in the protein folding, which is reflected in its loss of solubility and incorporation into inclusion bodies?

These residues are involved in an intricate and extended network of hydrogen bonds in the active site of the molecule. A disruption of this network (by mutations) must lead to decreased stability or inability to fold properly. A statement to this effect has been made stronger in the revised manuscript.

5. Some minor changes:

Page 6, Line 149: Ligands

Done.

Page 8, line 215: Michaelis-Menten

Done.

Page 8, line 222: Please include "CENTA" definition

Done.

Reviewer #2 (Remarks to the Author):

In their manuscript "The elusive *Rhizobium etli* L-asparaginase ... ", Loch et al. describe the structures, physicochemical characteristics and kinetics properties of asparaginase from *R. etli* (ReAV), native and its mutant forms. The authors employed quite a wide range of techniques in their studies, with X-ray crystallography being the dominant one. Although ReAV was described and to some extent characterized many years ago (Moreno-Enriquez et al., *J. Microbiol. Biotechnol.*, 2012, 22, 292-300, furthermore ME2012), this is the first report characterizing this enzyme structurally. As far as structural characterization is concerned, the presented material is of very high quality, in fact my overall impression is that it dominates the message of this report and as such would be more suitable for a journal focused on structural properties. Whereas this element can easily be corrected, I have a number of concerns regarding interpretation of results presented here.

1) In several instances the authors inform us about potential of ReAV for future utilization in the treatment of a human disease (leukemia). Unfortunately, I can't see this possibility. First, catalytic properties of ReAV are completely incompatible with those expected from anti-leukemia therapeutics. Values of K_M and k_{cat} (0.5 mM and 1.8 s⁻¹) are far from the desired values and it is extremely unlikely that this enzyme can be "optimized" to acceptable target values. Even these authors report that "optimal" activity of ReAV is observed at pH range 10-12 (very different from physiological conditions in human). Furthermore, similar data reported earlier for ReAV were vastly different (K_M and k_{cat} were 8.9 mM and 108 s⁻¹) (ME2012) and those authors reported optimal activity at 50°C (a potential problem in terms of therapeutic use). Taking into account that such fundamental characteristics as protease resistance (i.e. plasma half-life) and immunogenic reactivity are not evaluated here at all, a suggestion about a potential therapeutic use is a stretch. Additionally, the authors of this report don't discuss at all the great discrepancy between the results of their kinetic studies and those published earlier.

We understand the reservation of the Reviewer that the suggestion about ReAV as a potential antileukemic candidate was a long shot. We have, therefore, mitigated all such claims and only make a comment now that a study of asparaginases should be viewed in the context of the quest for new antileukemic drugs.

The issue of the apparent discrepancy between the kinetic parameters reported previously by ME2012 and in our original manuscript has been addressed above at a similar comment (1) from Reviewer #1. Briefly, in the revised manuscript we have redetermined the enzyme kinetic parameters using the same conditions as ME2012 and obtained much closer results ($K_M=15.8$ mM and $K_{cat}=278$ s⁻¹ at pH 7.5). The remaining differences may be attributed to different experimental method and to differences in protein preparation protocols. Specifically, the protein used by ME2012 contained a long (7.5 kDa) attachment (His-tag + antibody epitope + protease recognition sequence) at the N terminus which could interfere with the activity tests.

We want to explain (as clarified in the revised manuscript) that our kinetic parameters were determined at pH 7.5 (and for comparison at pH 8.0 and 9.0). In the revised manuscript, the results of the new ITC experiments conducted at pH 7.5, 8.0 and 9.0 are presented as new Fig. 4.

As remarked in our response to comment (1) by Reviewer #1, we were able to track down the source of the misestimated parameters presented in the original submission to insufficient substrate (relative to enzyme) concentration. At too low L-Asn concentration (due to solubility problems we had at that time), the enzyme could not reach V_{max} and the kinetic parameters were estimated incorrectly. We are very grateful to both Referees for the motivation to repeat the experiments and correct these parameters.

Regarding the temperature of optimal activity, our determined T_m of WT ReAV is 51°C (in agreement with other studies) and it is indeed difficult to expect that the enzyme can have optimal activity at 50°C. The ME2012 authors reported that "The enzyme showed a maximal activity at 50°C, but the optimal temperature of activity was 37°C". This issue is, however, not contentious anymore, as in the revised manuscript we no longer push the claim of therapeutic value of ReAV.

2) Glutaminase activity. I am a bit surprised that the authors did not report results from a simple assay using the Nessler reagent, monitoring progress of a reaction between ReAV and L-Gln, instead of employing ITC technique. Lack of the thermal effect during mixing (and a possible reaction) of ReAV and L-Gln does not indicate at all the absence of catalytic reaction. It may simply suggest that the reaction is primarily associated with entropic effects. In such a not uncommon case ITC is not the appropriate technique to study this reaction. It is quite possible that ReAV is devoid of L-glutaminase activity, but the results presented here are not demonstrating it conclusively.

Yes, we've tried to use the Nessler reaction to test for glutaminase activity but unfortunately this method turned out to be unsuitable. The reason is that L-Gln preparations (even of top grade and from several vendors) are contaminated and give positive Nessler reaction even without the addition of any enzyme. In this situation, we had to use the more cumbersome ITC method, and accept its shortcomings, as pointed out by the Reviewer. We have added an additional comment and reference about L-glutaminase testing in the revised manuscript. The problems with false positive Nessler reaction in the presence of L-Gln was reported many years ago by Wootton et al., (1976), however, it is often overlooked. As all our enzymatic assays were performed with rigorous inclusion of blank tests with no addition of the enzyme, we were able to detect the abnormalities. Importantly, we have never observed such anomalies with the use of L-asparagine as the substrate. As additional evidence of lack of glutaminase activity, we might cite the studies performed by Huerta-Zepeda et al. (1996) which suggested that *R. etli* L-asparaginases have no glutaminase activity, as in "R. etli grown on glutamine, and glutamate (...), asparaginase activity [measured as NH_3 release] was low". Finally, we note that a convincing proof of absence of glutaminase activity was presented by ME2012, who wrote "In the presence of L-glutamine in the assay, L-asparaginase activity was maintained to levels similar to control reaction, which confirmed non additive activity and an absence of glutaminase activity, because L-Gln does not compete with L-Asn for the active site of L-asparaginase from *R. etli*." Moreover, assuming that the act of releasing the NH_3 molecule to the aqueous solution is rather an enthalpy-producing event, very similar in the case of glutamine and asparagine, we concluded that it should generate a similar heat flow of the calorimetric baseline.

Wootton JC, Kavanagh JP, Baron AJ, Lovett MG. Re-investigation of the effects of L-glutamine and L-asparagine on the *Neurospora crassa* NADP-specific glutamate dehydrogenase. *Biochem J.* 1976 159 803-806. <https://pubmed.ncbi.nlm.nih.gov/137720/>

3) Appearance (identification) of Zn²⁺ cation is discussed in this report quite extensively, however, I would expect to see some comments on the differences between the reported results and those described earlier (ME2012). Also, another potential problem involves the determination of the identity of the Zn²⁺. Although ITC studies seem quite compelling (in terms of associated ΔH), how do the authors reconcile their results with those published earlier (ME2012), in which several divalent cations (including Zn²⁺) were found inhibitory for catalysis. Probably, much stronger support for identification of the nature of the cation could be achieved by anomalous diffraction experiments?

Yes, we have both X-ray fluorescence scans (now included as Supplementary Fig. S9) and anomalous maps confirming the presence and identity of zinc. We make it clear in the revised manuscript.

As discussed at a similar point (2) made by Reviewer #1, the experiments reported by ME2012 were carried out for a protein construct with His-tag attached, which would certainly interfere with any divalent metal binding studies. Also, the Zn²⁺ concentration used by ME2012 (1 mM) was high above the physiological level. Most importantly, the experiments reported by ME2012 were conducted in phosphate buffer, where many divalent metal cations precipitate as insoluble phosphates. This is certainly true for Zn²⁺. We cannot, therefore, treat the previous results with confidence. Nevertheless, we have conducted additional experiments (using Tris buffer) testing the influence of high zinc concentration, and report in the revised manuscript that at the highest physiological level of Zn²⁺, ReAV activity is reduced by just 29% in a non-specific manner.

Additional comments:

a: on few occasions the authors refer to "current classification of asparaginases" with the reference to their 2021 paper. I think that it is much more appropriate to refer to "recently proposed classification of asparaginases" as it is not yet universally accepted and certainly not "generally current classification"

We thank the Reviewer for this correction. We have changed our wording to "recently proposed classification of asparaginases", as suggested.

b: the reference 16 on page 4 (line 82) is incorrect

We thank the Reviewer for spotting this mistake. The references in this sentence have been rearranged and a new (correct) one has been added.

c: the authors acknowledge that the structure of ReAV is similar to some glutaminases. The rmsd value between ReAV and these enzymes, close to 1.2 Å, suggests very close similarity. Therefore, their claim of novelty of the ReAV structure needs to be considered much more carefully. Asparaginase and glutaminase activities are chemically closely-related and more compelling arguments need to be presented to claim the uniqueness of ReAV in this area.

Yes, we are also aware of the similarity of the asparaginase and glutaminase reactions; this is why we presented (to the best of our ability, vide supra) an experimental test (ITC) showing the absence of glutaminase activity. Moreover, we wish to point out that the asparaginase activity of ReAV has a well-established history in the literature, while no-one has so far discussed this protein as a glutaminase. Structurally, despite the relatively low rmsd value (1.48 Å, not 1.2 Å), there are

very important differences: the glutaminase homologs detected by DALI do not have metal-binding capability and dimerize differently.

In summary, this report represents a high-quality structural description of ReAV with a series of physicochemical and kinetic experiments characterizing this enzyme. Whereas I have an impression that structural details play a dominant role, I am not convinced about the interpretation of some physicochemical and kinetic data. Additionally, I do not believe in the postulated link between the results presented here and a potential therapeutic usefulness of this enzyme.

The interpretation of physicochemical and kinetic data has been given additional clarification, as explained in detail in the manuscript and above. The allusions to therapeutic usefulness of ReAV have been removed.

REVIEWER COMMENTS

Reviewer #1 (Remarks to the Author):

The paper entitled "The elusive *Rhizobium etli* L-asparaginase has a novel fold and peculiar active site" by Joanna I. Loch, Barbara Imiolczyk, Joanna Sliwiak, Anna Wantuch, Magdalena Bejger, Mirosław Gilski and Mariusz Jaskolski, has been revised according to the suggestions that were made.

The authors clarify the discrepancy with previous results and offer solid evidence of their results. They clearly discuss concerns raised regarding asparaginase enzymatic activity, kinetic parameters, and the role of zinc in catalysis.

I consider that the manuscript is ready for publication.

Reviewer #2 (Remarks to the Author):

I read the revised version of the manuscript by Loch et al. "The elusive *Rhizobium etli* L-asparaginase has a novel fold and peculiar active site" (manuscript number NCOMMS-21-25168B) and I believe that my concerns have been addressed satisfactorily. Therefore, I believe that the current version of the manuscript is (almost) acceptable for publication in *Nature Communications*. I still would like authors to discuss one additional aspect before the final acceptance. I consider it minor revision and leave it to your (editor) judgment.

Some of asparaginases were implicated previously to be allosteric enzymes (that includes some of type I bacterial asparaginases (ASnases) as well some examples of yeast orthologs). Since *Rhizobium etli* L-asparaginase is presented as the first example of a new class of ASnases, I believe that this point is important to look at, especially that cofactor (Zn^{2+}) is implied as important for catalysis. An additional (simple) analysis may be sufficient for this kind of assessment. Data from ITC (already available) could be fitted to the Hill equation (instead of Michaelis-Menten) to observe whether fit is improved and the Hill's coefficient differs significantly from value of 1. Ultimately, studies (again just a few experiments) of the enzymatic reaction at different Zn^{2+} concentrations may greatly illuminate role of this cation. I don't want to place this requirement as necessary for acceptance for the publication of manuscript but, I believe, it would greatly deepen characterization of this enzyme.

Finally, I am somewhat sceptic about author's explanation of failure during experiments with L-Gln using Nessler's reagent. We performed this assay over many years and never encountered a problem discussed by authors.

NCOMMS-21-25168B - Responses

Reviewer #1 (Remarks to the Author):

The paper entitled "The elusive *Rhizobium etli* L-asparaginase has a novel fold and peculiar active site" by Joanna I. Loch, Barbara Imiolczyk, Joanna Sliwiak, Anna Wantuch, Magdalena Bejger, Mirosław Gilski and Mariusz Jaskolski, has been revised according to the suggestions that were made.

The authors clarify the discrepancy with previous results and offer solid evidence of their results. They clearly discuss concerns raised regarding asparaginase enzymatic activity, kinetic parameters, and the role of zinc in catalysis.

I consider that the manuscript is ready for publication.

We thank the Reviewer for this highly positive opinion about our revised manuscript.

Reviewer #2 (Remarks to the Author):

I read the revised version of the manuscript by Loch et al. "The elusive *Rhizobium etli* L-asparaginase has a novel fold and peculiar active site" (manuscript number NCOMMS-21-25168B) and I believe that my concerns have been addressed satisfactorily. Therefore, I believe that the current version of the manuscript is (almost) acceptable for publication in Nature Communications.

We are grateful for this recommendation.

I still would like authors to discuss one additional aspect before the final acceptance. I consider it minor revision and leave it to your (editor) judgment.

Some of asparaginases were implicated previously to be allosteric enzymes (that includes some of type I bacterial asparaginases (ASnases) as well some examples of yeast orthologs). Since *Rhizobium etli* L-asparaginase is presented as the first example of a new class of ASnases, I believe that this point is important to look at, especially that cofactor (Zn^{2+}) is implied as important for catalysis.

We thank the Reviewer for bringing up the allosteric aspect. While we indeed found that ReAV is a metalloprotein with a Zn^{2+} cation bound in the active site, we also show that the binding of zinc in the active site does not affect the enzymatic activity. Similarly, the new Hill analysis with and without zinc gives the same result, excluding the role of Zn^{2+} as the allosteric effector.

An additional (simple) analysis may be sufficient for this kind of assessment. Data from ITC (already available) could be fitted to the Hill equation (instead of Michaelis-Menten) to observe whether fit is improved and the Hill's coefficient differs significantly from value of 1.

To assess the possibility of an allosteric effect, we have used, as suggested, the existing ITC titration data for fitting to the Hill equation (in addition to the Michaelis-Menten analysis already present in the manuscript). The result shows that there is a some degree of cooperativity, with the Hill coefficient calculated at $\sim 1.4-1.6$. There is, therefore, some allosteric effect, although the Hill analysis gives similar enzyme kinetics parameters. We have included a note about this extra

calculation in the Results section and present a short summary about the allosteric aspect in the Discussion.

Ultimately, studies (again just a few experiments) of the enzymatic reaction at different Zn²⁺ concentrations may greatly illuminate role of this cation. I don't want to place this requirement as necessary for acceptance for the publication of manuscript but, I believe, it would greatly deepen characterization of this enzyme.

As stated above (and indeed in the manuscript), the enzymatic activity of ReAV is practically the same with and without Zn²⁺ in the active site. The Hill analysis of the ITC data for native and Zn-free ReAV gives essentially the same kinetic parameters. Therefore, we can conclude that Zn²⁺ does not participate in catalysis and in allostery. As per a recommendation of the Editor, we have elected not to carry out additional experiments.

Finally, I am somewhat sceptic about author's explanation of failure during experiments with L-Gln using Nessler's reagent. We performed this assay over many years and never encountered a problem discussed by authors.

Our experience is consistent with similar observations reported by others, as supported by a suitable reference in our manuscript. It is possible that we may have been using different sources of L-Gln, but we think that we have done what is reasonably possible to test the highest-grade chemicals.